# Appetite Regulation, Growth Performances and Fish Quality Are Modulated by Alternative Dietary Protein Ingredients in Gilthead Sea Bream (*Sparus aurata*) Culture

**DOI:** 10.3390/ani11071919

**Published:** 2021-06-28

**Authors:** Lina Fernanda Pulido-Rodriguez, Gloriana Cardinaletti, Giulia Secci, Basilio Randazzo, Leonardo Bruni, Roberto Cerri, Ike Olivotto, Emilio Tibaldi, Giuliana Parisi

**Affiliations:** 1Department of Agriculture, Food, Environment and Forestry (DAGRI), University of Florence, 50144 Firenze, Italy; linafernanda.pulidorodriguez@unifi.it (L.F.P.-R.); giulia.secci@unifi.it (G.S.); leonardo.bruni@unifi.it (L.B.); 2Department of Agricultural, Food, Environmental and Animal Science, University of Udine, 33100 Udine, Italy; gloriana.cardinaletti@uniud.it (G.C.); roberto.cerri@uniud.it (R.C.); emilio.tibaldi@uniud.it (E.T.); 3Department of Life and Environmental Sciences, Polytechnic University of Marche, 60131 Ancona, Italy; b.randazzo@staff.univpm.it (B.R.); i.olivotto@univpm.it (I.O.)

**Keywords:** insect meal, microalgae, *Tisochrysis lutea*, *Tetraselmis suecica*, poultry by-product meal, red swamp crayfish meal

## Abstract

**Simple Summary:**

In a world dramatically harassed by climate changes and overexploitation of resources, the aquaculture expansion poses several challenges. Among the others, aquafeed formulations need to be rethought in a circular economy vision, avoiding food-feed competition, and possibly valorizing one or more ingredients for their functionality. In this context, the present trial showed that *Hermetia illucens* prepupae, poultry-by products, and red swamp crayfish meals can effectively replace a substantial proportion of vegetable proteins in the diet for gilthead seabream (*Sparus aurata*) without impairing fish growth and fillet quality. On the contrary, a blend of *Tisochrysis lutea* and *Tetraselmis suecica* negatively impacted fish growth and further studies are thus necessary to better valorize their natural content in pigments in fish farming.

**Abstract:**

By answering the need for increasing sustainability in aquaculture, the present study aimed to compare growth, gene expression involved in appetite regulation, physical characteristics, and chemical composition of *Sparus aurata* fed alternative protein sources. Fish were fed ten iso-proteic, iso-lipidic, and isoenergetic diets: a vegetable-based (CV) and a marine ingredient-rich (CF) diet were set as control diets. The others were prepared by replacing graded levels (10, 20 or 40%) of the vegetable proteins in the CV with proteins from a commercial defatted *Hermetia illucens* pupae meal (H), poultry by-product meal (PBM) singly (H10, H20, H40, P20, P40) or in combination (H10P30), red swamp crayfish meal (RC10) and from a blend (2:1, w:w) of *Tisochrysis lutea* and *Tetraselmis suecica* (MA10) dried biomasses. The increase in *ghre* gene expression observed in MA10 fed fish matched with increased feed intake and increased feed conversion ratio. Besides, the MA10 diet conferred a lighter aspect to the fish skin (*p* < 0.05) than the others. Overall, no detrimental effects of H, PBM, and RC meal included in the diets were observed, and fish fatty acid profile resulted as comparable among these groups and CV, thus demonstrating the possibility to introduce H, PBM, and RC in partial replacement of vegetable proteins in the diet for *Sparus aurata*.

## 1. Introduction

To date, aquaculture is the fastest growing farming sector in the world, answering the demand for safe and healthy food for a world population that will reach nearly 10 billion people by 2050 [1]. Recently, MacLeod et al. [2] calculated that global fish farming accounted for only approximately 0.49% of anthropogenic greenhouses gas emissions (263 Mt/53.5 Gt), thus resulting as a more virtuous way to produce high quality protein and lipid than other livestock sectors. However, by analysing the inputs contribution on the greenhouse gas value, a deep influence of feed ingredients emerged. Indeed, 39% of the total calculated aquaculture emissions derives from crop feed materials, and another 18% from fishmeal production, feed manufacturing and transport. In this regard, a challenge could be to turn this sector in a virtuous cycle, possibly using more sustainable and bioactive ingredients rather than conventional fish and soybean meals.

Both *Hermetia illucens* meal (H) and poultry by-product meal (PBM) are terrestrial animal proteins not directly destined for human consumption, hence they would help to raise this specific goal. It is widely established that the use of insects as a protein source has a number of advantages (high feed conversion efficiency, reduced land space and water consumption for their production), the most notable being the environmental benefits brought by the possibility to farm insects on bio-waste [3]. This could help to transform low value by-products into high-quality protein, which can be used as feed, and substantially contribute to reduce waste disposal and loss [4]. Analogously, processing poultry by-products into meal is a proper way to mitigate the environmental issues caused by this livestock sector and the overall price of feed, since PBM is cost-effective compared to other protein-rich feed sources. Besides, the production of 1 ton of proteins from H and PBM showed the lowest environmental impact on global warming, acidification, eutrophication, cumulative energy use and water use [5]. Early evidence on rainbow trout showed that moderate to high dietary levels of defatted H or PBM, replacing or complementing vegetable protein-rich ingredients in diets completely deprived of fish meal (FM), resulted in improved fish growth and gut health [6].

In addition, over the last years, microalgae have been studied in animal feed formulation. In particular, the dried marine microalgae biomass (DMB) is a source of lipids, especially n-3 long-chain polyunsaturated fatty acid (n-3 PUFA), with an essential amino acid profile similar to other conventional plant proteins. Likewise, its inclusion in the diet can provide nutrients and functional components such as vitamins, minerals, carotenoids, and natural antioxidants [7,8,9,10,11,12]. In aquaculture, DMB has been proposed as a potential protein and energy supplement that could replace marine ingredients and synthetic additives [7,10,11,13]. Despite the fact that the substitution of FM with increasing dietary inclusion levels of microalgae (such as *Tetraselmis suecica* and *Tisochrysis lutea*, singly or in combination) did not negatively affect the growth performance and feed conversion efficiency of European seabass [11,13,14,15], their high productive costs and uncertain availability still limits their application in aquafeed formulation.

Over the last decades, life below water and on land has been threatened by the proliferation of several alien species. In this regard, the case study of the red swamp crayfish (*Procrambarus clarkii*) needs to be mentioned. This freshwater crustacean is highly invasive as it preys on native species and damages streams, rivers and lakes because of its burrowing activity [16]. The fisheries management could be a strategy to keep this species under control; however, the biomass obtained by fishing should not be wasted, so that protein, lipid and pigment contained in the invasive species are not lost [17] by reducing red swamp crayfish into a meal (RCM); it can be used in aquafeed formulation supporting the fight against alien species and offering a new ingredient rich in proteins and pigments. Indeed, as underlined by Pulcini et al. [18], diets containing RCM (10%) and DBM (*Tisochrysis lutea* and *Tetraselmis suecica*, 10%) could provide carotenoids (50 mg g^−1^) [19], hence, they could be considered as functional ingredients, potentially acting as a natural source of pigments or antioxidants even at low concentrations, thus reducing the use of synthetic molecules in aquaculture.

Since a complex interplay of endocrine signals involved in appetite regulation between central (brain) and peripheral (intestine) systems exists, it is essential to analyse this complex cross-talk when testing new dietary ingredients [20,21,22,23]. Intestine is the main site for the production of ghrelin in fish, an orexigenic peptide produced by entero-endocrine cells [24]. Ghrelin acts as an appetite stimulator, inducing, at a central level, the expression of neuropeptide Y (*npy*), a more powerful appetite stimulator in the central system [25]. Moreover, there is a close relationship between *npy* expression and other endocrine signals in the brain. In particular, the cannabinoid system has been shown to have a central role in the regulation of appetite and exerts its function through the activation of different cannabinoid receptors, including the cannabinoid type 1 receptor (*cb1*) [26].

Standing on these premises, four different meals from terrestrial (*Hermetia illucens*, poultry by-product) or aquatic animals (red swamp crayfish) and microalgae (*Tisochrisis lutea* and *Tetraselmis suecica*) have been tested to partially substitute the vegetable proteins in complete feeds for gilthead sea bream (*Sparus aurata*). The present manuscript analyzed fish growth performances, health, welfare and food-quality attributes, measured by conventional (marketable traits, gas-chromatography) and innovative approaches (e.g., regulatory mechanisms of feeding behavior in fish).

## 2. Materials and Methods

### 2.1. Ethical Statement and Experimental Diets

The feeding trial was carried out at the aquaculture facilities of the Department of Agricultural, Food, Environmental and Animal Sciences of the University of Udine, according to the European Directive 2010/63/EU of the European Parliament and of the Council of the European Union on the protection of animals used for scientific purposes. The experimental protocol was approved by the Ethics Committee of the University of Udine and authorized by the Italian Ministry of Health (n. 290/2019-PR).

Ten test diets were formulated to be grossly iso-proteic (45%), iso-lipidic (20%) and isoenergetic (22 MJ kg^−1^ gross energy). A diet rich in plant-derived ingredients (named CV) was designed to obtain a 90:10 weight ratio between vegetable and marine proteins and a 67:33 weight ratio between vegetable and fish lipids, as calculated from the crude protein and lipid contribution to the whole diet of all marine and plant-based dietary ingredients. A diet rich in fish meal (CF) was formulated in the opposite way, i.e., to obtain a 10:90 weight ratio between vegetable and marine proteins and a 33:67 weight ratio between vegetable and fish lipids. The remaining diets were prepared replacing graded levels (10, 20 or 40%) of crude protein from the mixture of vegetable protein sources of the CV diet by crude protein from a commercial defatted *Hermetia illucens* pupae meal (H10, H20, H40), poultry by-product meal (P20, P40) singly or in combination (H10P30, plant proteins were replaced by 10% protein from H meals and 30% protein from PBM), red swamp crayfish meal (RC10) and from a blend (2:1 w:w) of the dried biomass of two marine microalgae (MA-*Tisochrysis lutea* and *Tetraselmis suecica*, MA10), respectively, while maintaining the same 67:33 vegetable to fish lipid ratio as in the CV diet.

All diets were manufactured by SPAROS Lda (Olhão, Portugal) by extrusion in two pellet sizes (3 and 5 mm) and stored at room temperature (+4 °C) until they were administrated. The ingredient composition and chemical composition of the test diets are shown in Table 1. The fatty acid (FA) profile of the test diets is reported as Appendix A.

### 2.2. Fish Rearing and Sampling

The experiment utilized juvenile gilthead seabream (initial mean body weight 48.8 ± 8.8 g) from a resident stock. Fish were randomly divided to form 30 groups, each including 18 specimens, which were kept in cylindrical fiberglass tanks with a capacity of 300 L each. The tanks were part of an indoor, marine recirculating aquaculture system (RAS) equipped with a mechanical and biological filter, a protein skimmer and a UV lamp (Scubla srl, Remanzacco, Udine, Italy), which ensured optimal water quality for fish (water temperature, 23.6 ± 0.70 °C; salinity, 30 ± 1.4 g L^−1^; dissolved oxygen, 6.1 ± 0.38 mg L^−1^; pH, 8.0 ± 0.1; Total Ammonia Nitrogen < 0.015 mg L^−1^; N-NO_2_, 0.10 ± 0.03 mg L^−1^). During the feeding trial, fish were kept under constant day length and light intensity (12 h per day at 400 lux) provided by fluorescent light tubes. Fish groups were left to adapt to the culture conditions over two weeks before being randomly assigned in triplicate to the 10 diets. Fish were fed the test diets six days a week over 21 weeks. The diets were delivered by belt feeders in two daily meals (8:00 a.m. and 4:00 p.m.) slightly in excess to assure fish satiety. Satiety was attained by distributing a daily feed amount adjusted to exceed the intake of the previous day so as to obtain feed residues after each meal. To this end, the outlet of each tank was fitted with an apparatus for recovering uneaten feed pellets shortly after being released by the feeder. Feed amounts distributed to each tank were recorded daily and uneaten feed items were recovered, dried, and weighed to estimate actual feed intake. After 12 weeks of the trial, fish density in each tank was reduced to ten fish.

At the end of the feeding trial, all fish were weighed in bulk after 24 h fasting and under moderate anaesthesia with MS-222 (PHARMAQ Ltd., Fordingbridge, Hampshire, UK). The following parameters were calculated:Feed Intake (g/kg/ABW/d) = feed intake per tank/[(initial biomass + final biomass)/2)/days](1)
Specific Growth Rate (SGR) = 100 × [(ln final body weight-ln initial body weight)/days];(2)
Feed Conversion Ratio (FCR) = feed intake per tank/weight gain per tank.(3)

Subsequently, all fish were sacrificed with an overdose (300 ppm) of the same anesthetic and subjected to the analyses detailed below.

### 2.3. Gene Expression Analyses

#### 2.3.1. RNA Extraction and cDNA Synthesis

Samples were prepared according to Olivotto et al. [27,28]. Briefly, Total RNA was extracted from medium intestine and brain samples (n = 9 for each experimental group) using RNAzol^®^ RT reagent (Sigma-Aldrich^®^, R4533, Milan, Italy) and following the manufacturer’s instructions. RNA concentration and integrity were analysed using NanoPhotometer^®^ P-Class (Implen, Munich, Germany) and Gel Red™ staining of 28S and 18S ribosomal RNA bands on 1% agarose gel, respectively. After extraction, complementary DNA (cDNA) was synthesized from 1 μg of total RNA with the LunaScript RT SuperMix Kit (New England Biolabs, Ipswich, MA, USA), following the manufacturer’s instructions, diluted 1:10 in RNase-DNase free water and stored at −20 °C until use. An aliquot of cDNA was used to check primer pair specificity.

#### 2.3.2. Real Time PCR

The mRNA levels of selected genes involved in appetite in intestine and brain were assessed. Specifically, ghrelin (*ghre*) expression was analysed in medium intestine; cannabinoid receptor (*cb1*) and neuropeptide Y (*npy*) expression was analysed in the brain. The primers sequences were retrieved from NCBI (http://www.ncbi.nlm.nih.gov/ accessed on 26 January 2021) and are summarized in Table 2. Amplification products were sequenced, and homology was verified. Negative controls revealed no amplification product, and no primer-dimer formation was found in control templates.

PCRs were performed according to Piccinetti et al. [29] and Vargas-Abúndez et al. [30] in an iQ5 iCycler thermal cycler (Bio-Rad, Milan, Italy) and each sample was analysed via RT-qPCR in triplicate. Reactions were set on a 96-well plate by mixing, for each sample, 1 μL cDNA diluted 1:20.5 μL of 2× concentrated iQ™ Sybr Green (Bio-Rad, Milan, Italy) as the fluorescent intercalating agent, 0.3 μM forward primer, and 0.3 μM reverse primer. The thermal profile for all reactions was 3 min at 95 °C, followed by 45 cycles of 20 s at 95 °C, 20 s at 60 °C, and 20 s at 72 °C. Fluorescent signal was detected at the end of each cycle and the melting curve analysis was performed to confirm that only one PCR product was present in these reactions.

For the gene expression relative quantification, beta-actin (*β-actin*) and ribosomal protein S18 (*rps18*) RNA were used as housekeeping genes to standardize the results. Data were analysed using the iQ5 optical system software version 2.0, including Genex Macro iQ5 Conversion and Genex Macro iQ5 files (all from Bio-Rad). Modification of gene expression was reported with respect to all the groups. Primers were used at a final concentration of 10 pmol μL^−1^.

### 2.4. Physical and Chemical Analyses on Fillets

#### 2.4.1. Marketable Indexes and Physical Analyses

Ten fish per dietary treatment were thawed overnight at +1 °C before being analysed. Firstly, the fish were individually weighed (as eviscerated), total and muscular lengths were measured, and the measurement of skin colour was carried out; then, the fish were dissected. The following parameters were calculated:Condition Factor, K (%) = [(body weight (g)/total length (cm)^3^] × 100(4)
Fillet Yield, FY (%) = [(fillet with skin weight (g)/body weight (g)] × 100(5)
Hepatosomatic Index, HSI = [(liver weight (g)/total body weight (g)] × 100.(6)

The colour of skin and fillets was measured on triplicate positions (cranial, medial and caudal) on both fish sides with a CHROMA METER CR-200 (Konica Minolta, Chiyoda, Japan) following the CIELab system [31], thus recording *L** (lightness), *a** (redness index) and *b** (yellowness index) colour parameters.

The values of pH and of maximum shear force (texture) parameters were registered. The pH value was measured on triplicate fillet positions (cranial, medial, and caudal) by a pH-meter SevenGo SG2™ (Mettler-Toledo, Schwerzenbach, Switzerland). Texture was assessed as the maximum shear force value obtained utilising the Warner-Bratzler shear blade (width of 7 cm) by a Zwick Roell^®^ 109 texturometer (Zwick Roell, Ulm, Germany), equipped with a 1 kN load cell, setting the crosshead speed at 30 mm min^−1^. Afterwards, fillets were skinned, homogenized, and utilized to determine the chemical composition, as described below.

#### 2.4.2. Chemical Composition and Fatty Acid Profile

Water content was determined using 2 g of sample by heating at 105 °C until constant weight [32]. Total nitrogen was determined using the Kjeldahl procedure (Kjeltec, 1035 Analyzer, Foss, Hilleroed, Denmark) and converted to crude protein by multiplying by 6.25 [32]. Ash was determined as the remnant weight after calcination of a 5 g sample at 550 °C for 3 h [32]. The results were expressed as g 100 g^−1^ product. The total lipids of the samples were obtained according to Folch et al. [33], then they were gravimetrically quantified. The fatty acids (FA) were determined in the lipid extract after transesterification to methyl esters (FAME), using a base-catalysed trans-esterification [34]. The FA composition was determined by gas-chromatography (GC) using a Varian GC 430 gas chromatograph (Varian Inc., Palo Alto, CA, USA), equipped with a flame ionization detector (FID) and a Supelco Omegawax™ 320 m capillary column (Supelco, Bellefonte, PA, USA). The condition of the GC analysis was set as previously mentioned [35]. Chromatograms were recorded with the Galaxie Chromatography Data System 1.9.302.952 (Varian Inc., Palo Alto, CA, USA). FAs were identified by comparing the FAME retention time with those of the Supelco 37 component FAME mix standard (Supelco, Bellefonte, PA, USA) and quantified through calibration curves, using tricosanoic acid (C23:0) (Supelco, Bellefonte, PA, USA) as internal standard.

#### 2.4.3. Fillet Oxidative Status

Primary and secondary oxidative products were quantified in homogenized fillets as conjugated dienes (CD) and thiobarbituric acid reactive substances (TBARS), respectively. The CD were quantified in 0.5 µL of lipid extract dissolved in 3 mL of pure hexane. The absorbance at 232 nm (50 Scan spectrophotometer Varian, equipped with a Cary Win UV Software) was determined, and the mmol hydroperoxides kg^−1^ sample were calculated using a molar extinction coefficient of 29,000 mL mmol^−1^ × cm [36]. Finally, 2 g of homogenized fillets were utilized to determine the secondary lipid oxidation products (TBARS). The TBARS content was measured using the colorimetric method described by Secci et al. [35], using trichloroacetic acid (5%) as solvent and then adding TBA 0.02 M. After 40 min of incubation at 97 °C, the oxidation products were quantified with reference to calibrations curves of TEP (1,1,3,3-tetra-ethoxypropane) in 5% (*w*/*v*) TCA (from 0.2 to 3.1 mmol L^−1^). The absorbance at 532 nm was read with a 50 Scan spectrophotometer equipped with Cary Win UV software (Varian Inc., Palo Alto, CA, USA).

### 2.5. Statistical Analyses

Growth performance data were checked for normal distribution and homogeneity of variance before analysis by using the SPSS-PC release 17.0 (SPSS Inc., Chicago, IL, USA). The data related to the marketable indexes, physical and chemical analyses were subjected to one-way analysis of variance (ANOVA) using the PROC GLM of SAS/STAT Software, Version 9.4 [37], followed by Tukey’s multiple-comparison test to assess significant differences among the groups. Significance was set at *p* < 0.05 and all the results are presented as mean and root mean square error (RMSE). The statistical software package Prism5 (Graphpad Software, La Jolla, CA, USA) was used for genetic analyses.

## 3. Results

### 3.1. Growth Performance

The growth performance, feed intake and conversion ratio of gilthead sea bream fed the experimental diets over 21 weeks are shown in Table 3. Fish fed diet MA10 resulted in the highest feed intake and in the worst growth performance when compared to both control diets and the other dietary treatments (*p* < 0.05). A slightly higher feed consumption was observed with diets CF and RC10 relative to treatments H10, H20, H10P30 and P40, even though this did not result in a parallel improvement of growth response. As a consequence of reduced growth and increased feed intake, fish fed diet MA10 exhibited the worst feed conversion ratio (*p* < 0.05) while diets CF and RC10 resulted in intermediate feed conversion ratio values between the former one and those attained with the other dietary treatments (*p* < 0.05).

### 3.2. Gene Expression

The analysis of *ghre* gene expression in the medium intestine and gene expression of *cb1* and *npy* in the brain of fish fed the different diets are shown in Figure 1. Fish fed diet MA10 displayed a significant increase in *ghre* mRNA levels in medium intestine compared to that of fish subjected to the other dietary treatments (*p* < 0.05). In the brain of breams given diets H10P30 and RC10, gene expression of *cb1* and *npy* was found significantly higher compared to those of fish fed the other diets (*p* < 0.05).

### 3.3. Analyses on Fillets

The marketable indexes and physical characteristics of the fillets from gilthead sea bream were not significantly affected by dietary treatments, except for the colour (Table 4). The skin lightness index (*L**) of the P20 fish was lower than that of the MA10 ones (*p* < 0.05), while all the other groups had intermediate values. A pronounced yellow index (*b**) was registered for skin of fish fed the H40 diet (*p* < 0.05) that significantly differed from CF, P40, and H10P30.

Table 5 shows the results of the chemical composition and FA profile of the gilthead sea bream fillets. The chemical composition of fresh fillets did not differ between the dietary groups (*p* > 0.05) whilst the fillet FA profile was deeply affected by the diet (*p* < 0.001). Oleic (C18:1n-9), palmitic (C16:0), and linoleic (C18:2n-6) acids were the major FAs in all the groups, representing on average 31.1, 15.1 and 14.9% in each dietary treatment, respectively. Overall, monounsaturated fatty acids were abundantly present in all the dietary treatments (from 38.33 to 40.94%) showing values significantly higher in P20 and P40 dietary groups, while the CF group presented the lowest values (*p* < 0.05). In contrast, the saturated fatty acid incidence was higher in the CF group, and significantly lower in the diets including vegetable source of protein (CV and MA10) and in RC10 diet (*p* < 0.05). On the other hand, C18:2n-6 was more abundant in fillets from fish fed MA10 than in fillets from those fed graded level of *Hermetia* and PBM (*p* < 0.05), and from CF group, that showed the lowest value (*p* < 0.05). Since C18:2n-6 is the most representative n-6 PUFA, the difference previously described for linoleic acid was found for the percentages of n-6 PUFA sum. The C18:3n-3 (α-linolenic acid) represented an average 7.6% of the lipid composition of the fillets with those of fish fed the vegetal control and the MA10 diets resulting in the highest values (8.46% and 8.37%, respectively). The proportions of DHA (C22:6n-3) and EPA (C20:5n-3) were significantly higher in the fillet of fish fed diet CF (*p* < 0.05) that also resulted in elevated n-3/n-6 PUFA ratio relative to the other groups (*p* < 0.05).

The results of lipid oxidation (Table 6) revealed that the diets significantly affected the conjugated dienes content of the fillets. The fillets of fish fed diet CF were the most prone to lipid oxidation, while those of fish fed the vegetal control and MA10 diets appeared the most stable (*p* < 0.05). On the other hand, the degree of lipid oxidation, in terms of TBARS, was not affected by dietary treatments (*p* > 0.05).

## 4. Discussion

The aquaculture industry, in its attempt to improve the sustainability of production and maintaining a vision of a circular economy, has directed various strategies to replace marine ingredients with protein-rich plant derivatives. However, other categories of unconventional protein sources (the third-generation ingredients) have recently been tested. *Hermetia illucens* meals have been studied in a previous experiment on gilthead sea bream, where its inclusion of up to 40% in diets mainly based on conventional vegetable or marine ingredients did not show negative effects on growth performance and feed efficiency [38]. Similarly, the growth performance of gilthead sea bream was not impaired when poultry by-product meal was included at 25, 36, or 40% in the diets to replace fish meal [38,39,40]. The results of the present study are in agreement with the previous ones, showing that optimal zootechnical performance were achieved irrespective of the inclusion levels of *Hermetia illucens* and poultry by-product meals, singly or in combination.

Changes in growth response to varying protein sources in the diet could primarily reflect differences in feed consumption. Feed intake is controlled by a dual component, including a short-term (meal to meal) regulation of feed intake mediated by central and peripheral signals and a long-term feedback regulation (days to months) modulated by body energy stores and food availability over prolonged periods [41]. The two mechanisms work together to integrate energy intake and expenditure to ensure the maintenance of energy balance [42]. Because of the crosstalk between intestine and brain, analysing appetite-related genes in the central nervous system and in the intestine is necessary. On this regard, in vertebrates (fish included) at central nervous system level *npy* and *cb1* are important molecules involved in enhancing the appetite stimulus and body weight [23,43].

The results obtained by the *npy* and *cb1* gene expression analysis in the present study underlined that none of the dietary treatments seemed to depress central neuro-endocrine mechanisms involved in appetite *stimulus*. This result may be related to a possible long-term central adaptative feed-back response by the fish to 21 weeks feeding trial.

On the contrary, *ghre* gene expression, which is often related to a short term (meal to meal) regulation, showed interesting results which appeared well related to the feed intake data. The increased *ghre* gene expression observed in the medium intestine of fish fed diet MA10 relative to the other dietary groups, was consistent with the zootechnical results (increased feed intake and worst feed conversion ratio) and may indicate a local response to the microalgae dietary inclusion. According to previous studies, the addition of the microalgae blend in the diet, particularly *Tetrasemis*, probably depressed diet digestibility. Poor digestibility of intact cell *Tetraselmis* biomass has recently been observed in European sea bass [44]. Depressed dietary nutrient and gross energy apparent digestibility was also reported in previous studies with E. sea bass fed diets including a variable proportion of intact-cell dried biomass of *T. suecica* alone or blended with *T. lutea* [13,14]. This seems a consequence of a recalcitrant cell wall which makes cell contents of *T. suecica* poorly accessible to digestive enzymes. As a consequence, in the present study, fish fed diet MA10 exhibited the greater (compensatory) feed consumption which, however, was not sufficient to ensure growth performances comparable to those attained by the other dietary groups.

These data agreed with previous studies, which highlighted that only low (e.g., <10%) dietary inclusion levels of dried intact-cell microalgae biomass are tolerable without negatively impacting growth performance of fish [45,46,47,48].

Colour is one of the most common physical attributes considered while assessing fish quality. As commonly accepted, the economic value of farmed gilthead sea bream grows directly with an increase in skin pigmentation, especially in the forefront area between the eyes [49]. Because of this, researchers have been focusing on different dietary interventions to enhance skin colour using a variety of pigment sources, such as microalgae [49,50,51,52,53], synthetic astaxanthin [51] and vegetable sources [54,55]. Despite the fact that the use of crustaceans as an active colourant has been previously investigated, the information on the use of *Procambarus clarkii* is limited to the only study performed on *Pagrus pagrus* by García et al. [56]. Overall, the nature of the administered source, the type and content of the dietary carotenoids affect the magnitude of colour modification with major emphasis given on the correlation between total carotenoid concentration in the feed and yellowish pigmentation of the fish skin [18]. In this regard, Pulcini et al. [18] previously published the data on total carotenoid content of the same diets here administered. Briefly, the MA10 had the highest carotenoid concentration, followed by the vegetable control and RCM diets, the insect included diets (H10, H20 and H40), whilst the P40 and H10P30 showed the lowest amounts. The authors showed that the number of yellow pixels determined in the lateral side of fish body varied accordingly, and it was possible to distinguish three clusters: one with fish fed MA10, RC10 and P20, one including the CV fish, and the third cluster with H20, H40 and H10P30 gilthead sea bream. The different analytical methods adopted to describe the colour (i.e., image analysis against colorimetric punctiform analysis) might be responsible for the inconsistencies between Pulcini et al.’s data and the present ones, where the skin yellowness seemed improved in the H40 dietary group. However, the present results agree with Ribeiro et al. [52] who found that *Phaeodactylum tricornutum* biomass included at 2.5% in the diet for gilthead sea bream led to significant lighter operculum and ventral skin than the control diet. On the contrary, the fish fillets were unaffected by the dietary intervention and this result agreed with Gouveia et al. [49] who underlined that, despite the astaxanthin dietary source provided by *Chlorella vulgaris* (Chlorophyta, Volvocales), muscle carotenoids amounted to less than 1 mg kg^−1^ and no significant colour differences emerged in the gilthead sea bream fillets.

A strong relationship exists between the composition and properties of the diet offered to fish and the nutritional traits of the fish body, that is, the final product [11,13,38,56,57,58]. According to this, the changes in the lipid composition of the diets were reflected in the FA profile of the fillets. It was expected that the substitution of marine ingredients in diets with protein and lipid sources of animal or vegetal origin would increase the saturated, monounsaturated and n-6 PUFA content [11,59]. For instance, the lipid composition of *Hermetia illucens*, which is characterized by high contents of saturated fatty acid and mainly in lauric acid [58,60,61,62], caused the increase in the C12:0 values observed in the fillets of the groups fed H10, H20, H40 and HI10P30. Dietary deprivation of marine ingredient led the expected n-3 PUFA reduction. However, while comparing the n-3 PUFA content of the experimental diets and the fillets, it emerged that H10, P40 and RC10 were partially able to counteract such a decrease, since their n-3 PUFA content increased on average 2.23% in comparison to the corresponding diets. In addition, the present results highlight that DHA increased on average 36.8% in the fillets of fish fed on unconventional ingredients with respect to CV and CF groups whose DHA content increased by 30.5 and 23.13%, respectively. These results suggested that n-3 PUFA were selectively deposited and retained in the muscle tissue of fish fed diets containing alternative ingredients (individually or in combination and at different inclusion levels). For instance, the plant ingredients (particularly soybean meals) can reduce lipid absorption and retention in fish tissues [63,64,65], and their partial replacement with *Hermetia illucens* and poultry by-product meals in graded levels (20% and 40%) in the feed for *Sparus aurata* has been associated to an improved lipid absorption [38]. The selective deposition of DHA in the muscle could be regulated by the elongase and desaturase enzymatic activity on C18:3n-3. Indeed, the genes coding for these enzymes are generally up-regulated by increasing dietary α-linolenic acid or decreasing n-3 PUFA content [57]. This could explain the pattern here obtained for P40 fish, since the α-linolenic acid relative abundance in feed was the lowest among all the other experimental diets. Although marine fish exhibit a limited elongase and desaturase activities [57], a recent study showed that diets low in n-3 PUFA up-regulated the gene expression of desaturase and elongase in *Sparus aurata* [66], thus supporting the present findings about FA profile of the fish fed diets totally deprived of fish meal. Finally, García-Romero et al. [67] recorded the reduction in EPA in the fillet of fish fed a diet that included up to 10% of *Procambarus clarkii* meal, while the concentration of DHA in muscle tissue remained above the value contained in the diet, triggering the *Pagrus pagrus* preference to use EPA for energy purposes while DHA was selectively deposited, as shown in the present study. Concerning the microalgae blend, it did not enhance the PUFA profile of fish fillet probably because of the low dry matter and gross energy apparent digestibility previously observed [13]. However, the overall PUFA profile, synthesized in the n-3/n-6 ratio, highlighted that the alternative dietary groups did not suffer strong negative effects when compared to the vegetable control diet.

The partial substitution of the vegetable ingredients with non-conventional proteins from animal and microbial origin decreased the oxidation rate, as expressed by conjugated dienes content analysis. Despite the fact that the decrease in lipid oxidation might be related to the low PUFA contents in muscle tissue, which are very prone to oxidation, in the present study the fillets from fish fed alternative diets maintained higher mean values of the total PUFA fraction (36.4 ± 1.15 g 100 g^−1^ total FAME) compared to the fish meal control diet (35.4 ± 1.15 g 100 g^−1^ total FAME). Li et al. [68] indicated that the use of *Hermetia illucens* in the diet for Jian carp (*Cyprinus carpio* var. Jian) could increase the activity of serum catalase, which could improve the antioxidant defense of fish. Similarly, García-Romero et al. [67] reported that the presence of antioxidant components present in freshwater crab meals protects FAs from lipid oxidation in *Pagrus pagrus* fillets. On the other hand, the low rate of lipid oxidation observed in the fillets from MA10 group could be attributed to the bioactive compounds present in the dry biomass of microalgae [13,69].

## 5. Conclusions

The present study showed that meals from terrestrial animals (*Hermetia illucens* or poultry by-product) or from aquatic animals (red swamp crayfish) appear to be promising protein sources for the replacement of vegetal proteins in complete feeds for *Sparus aurata*. On the contrary, a low inclusion level of a blend of dried biomass of intact-cell marine microalgae (*Tisochrisis lutea* and *Tetraselmis suecica*) resulted in depressed growth and the worst feed conversion ratio which seemed attributable to reduced diet digestibility. However, the final qualitative traits of the fillets from fish fed the diets including all unconventional protein sources were not adversely affected, and a nutritious final product for consumers was guaranteed.

## Figures and Tables

**Figure 1 animals-11-01919-f001:**
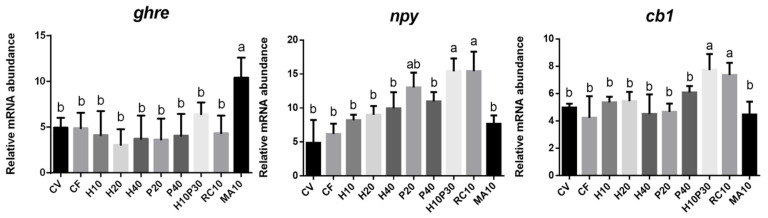
Gene expression of *ghre* in the medium intestine and *npy* and *cb1* in the brain of fish fed different diets. a, b: different superscript letters indicate significant difference among groups (*p* < 0.05).

**Table 1 animals-11-01919-t001:** Ingredient (g 100 g^−1^) and chemical composition (% as feed) of the experimental diets.

	CV	CF	H10	H20	H40	P20	P40	H10P30	RC10	MA10
**Ingredient composition**										
Fish meal ^1^		14.0								
Fish meal *(trimmings)* ^2^		40.0								
Feeding stimulants ^3^	5.5	5.5	5.5	5.5	5.5	5.5	5.5	5.5	5.5	5.5
Veg.-protein mix ^4^	69	-	60.5	52.6	36.6	52.5	35.4	35.4	58.8	58.3
*Hermetia* meal ^5^	-	-	8.10	16.2	32.4	-	-	8.1	-	-
PBM ^6^	-	-	-	-	-	13.8	27.5	20.6	-	-
RC meal ^7^	-	-	-	-	-	-	-	-	10.1	-
Microalgae mix ^8^	-	-	-	-	-	-	-	-	-	11.6
Wheat meal *	0.4	3.0	0.6	1.6	4.5	3.0	5.6	5.5	0.4	-
Whole pea *	3.0	20.5	4.8	5.8	6.0	6.2	9.0	8.8	4.1	4.0
Fish oil ^9^	6.2	8.6	6.2	6.2	6.2	6.2	6.2	6.2	6.2	6.2
Veg. oil mix ^10^	11.4	6.5	10.0	8.4	5.4	9.8	8.2	7.4	10.8	10.5
Vit. & Min. Premix ^11^	0.3	0.3	0.3	0.3	0.3	0.3	0.3	0.3	0.3	0.3
Choline HCL	0.1	0.1	0.1	0.1	0.1	0.1	0.1	0.1	0.1	0.1
Sodium phosphate	1.6	-	1.5	1.2	1.0	0.7	0.3	0.2	1.5	1.3
L-Lysine ^12^	0.5	-	0.5	0.2	0.2	0.1	0.1	0.1	0.3	0.3
DL-Methionine ^13^	0.5	-	0.4	0.4	0.3	0.3	0.3	0.3	0.4	0.4
Celite	1.5	1.5	1.5	1.5	1.5	1.5	1.5	1.5	1.5	1.5
**Chemical composition**										
Moisture	6.5	8.2	4.2	6.0	4.5	7.1	7.1	8.6	6.1	3.5
Crude protein (N × 6.25)	45.1	45.4	45.5	45.3	45.2	45.1	45.1	45.2	45.4	45.2
Total lipids	20.4	20.3	20.2	20.2	20.4	20.5	20.3	20.4	20.1	20.4
Ash	5.8	12.4	6.7	6.6	6.6	7.1	7.8	7.7	8.9	6.9
Chitin ^#^	0.02	0.02	0.40	0.76	1.51	0.02	0.02	0.39	0.73	0.02

^1^ Fish meal: Pesquera Diamante Peru (65.3%, crude protein CP; 11.5%, crude fat CF). ^2^ Fish meal trimmings: Conresa 60, Conserveros Reunidos S.A. Spain (59.6% CP; 8.9%, CF). ^3^ Feeding stimulants g/100 diet: fish protein concentrate CPSP90-Sopropeche, France (82.6% CP), 3.5; Squid meal (80.3% CP), 2.0. ^4^ Vegetable-protein sources mixture (% composition): dehulled, toasted soybean meal, 39; soy protein concentrate-Soycomil, 20; maize gluten, 18; wheat gluten, 15, rapeseed meal, 8. ^5^ ProteinX™, Protix, Dongen, Netherlands (CP, 55.4%; CF, 20.8% as fed) ^6^ Poultry by-product meal from Azienda Agricola Tre Valli; Verona, Italy (CP, 65.6%; CF, 14.8% as fed). ^7^ Red swamp crayfish, *Procambarus clarkii* (CP, 44.4%; CF, 8.7% as fed). ^8^ Dry microalgae biomass mixture from (% composition): *Tisochrysis lutea* meal, 63.8 (CP, 40.7%; CF, 10.9% as fed); *Tetraselmis suecica* meal, 36.2% (CP, 35.9%; CF, 8.9% as fed). ^9^ Fish oil: Sopropêche, Boulogne sur Mer, France. ^10^ Vegetable oil mixture, % composition: rapeseed oil, 56; linseed oil, 26; palm oil, 18. ^11^ Vitamin and mineral supplement (per kg of premix): Vit. A, 2,000,000 IU; Vit D3, 200,000 IU; Vit. E, 30,000 mg; Vit. K3, 2500 mg; Vit.B1, 3000 mg; Vit. B2, 3000 mg: Vit B3, 20,000 mg; Vit. B5, 10,000 mg; Vit B6, 2000 mg, Vit. B9, 1500 mg; Vit. B12, 10 mg; Biotin, 300 mg; Stay C^®^, 90,000 mg; Inositol, 200,000 mg; Cu, 900 mg; Fe, 6000 mg; I, 400 mg; Se, 40 mg; Zn, 7500 mg. ^12^ L-lysine, 99%; Ajinomoto EUROLYSINE S.A.S; France. ^13^ DL-Methionine: 99%; EVONIK Nutrition & Care GmbH, Germany. * Wherever not specified, the ingredients composing the diets were obtained from Sparos Lda. ^#^ Estimated based on chitin content supplied by feed ingredients (squid meal, 0.9%; *Hermetia illucens* meal, 4.69%; *Procambarus clarkii* meal, 7.2%).

**Table 2 animals-11-01919-t002:** Oligonucleotide primers, annealing temperature (A.T.) and location (Gene Bank Accession Number) of each gene investigated in this study. *hk*: housekeeping genes.

Gene Name	Primer Sequence		A.T. (°C)	Gene Bank ID
	Forward	Reverse		
*ghre*	GGAAAGTCTTCCAGGGTCGG	CGCATAGTCCTCTTCTGTCATGGAG	59	MK089519.1
*cb1*	GCTGGGCTGGAACTGTAAAC	TTCCACAGGATGTATATGTAGGC	60	EF051620.1
*npy*	GGAGCTGGCCAAGTACTACTCA	GAGACCAGCGTGTCCAGAAT	60	XM_030411288.1
*β-actin (hk)*	TCCTGCGGAATCCATGAGA	GACGTCGCACTTCATGATGCT	57	X89920.1
*rps18 (hk)*	AGGGTGTTGGCAGACGTTAC	CTTCTGCCTGTTGAGGAACC	57	AM490061.1

**Table 3 animals-11-01919-t003:** Final weight (FW, g), feed intake (FI, g/kg ABW/d), specific growth rate (SGR, g/kg ABW/d) and feed conversion ratio (FCR) of gilthead sea bream (*Sparus aurata*) fed the experimental diets over 21 weeks.

	CV	CF	H10	H20	H40	P20	P40	H10P30	RC10	MA10	*p* Value	RMSE
FW	327.2 ^a^	327.5 ^a^	334.1 ^a^	349.7 ^a^	343.7 ^a^	335.6 ^a^	342.6 ^a^	349.8 ^a^	330.4 ^a^	302.8 ^b^	0.035	115.58
FI	11.7 ^bc^	12.2 ^b^	11.3 ^c^	11.5 ^c^	11.7 ^bc^	11.9 ^bc^	11.7 ^bc^	11.6 ^c^	12.1 ^b^	13.6 ^a^	0.006	0.07
SGR	1.32 ^ab^	1.31 ^b^	1.33 ^ab^	1.36 ^a^	1.35 ^a^	1.33 ^ab^	1.34 ^ab^	1.36 ^a^	1.32 ^ab^	1.26 ^c^	0.009	0.0003
FCR	1.18 ^a^	1.25 ^b^	1.15 ^a^	1.16 ^a^	1.15 ^a^	1.15 ^a^	1.16 ^a^	1.14 ^a^	1.24 ^b^	1.39 ^c^	<0.001	0.0003

CV, vegetable control; CF, fish meal control; H, *Hermetia illucens*; P, poultry by-product; RC, red swamp crayfish; MA, microalgae dried biomass. RMSE: root mean square error. ^a,b,c^: different superscript letters indicate significant difference among groups (*p* < 0.05).

**Table 4 animals-11-01919-t004:** Marketable indexes and physical characteristics of the fillets from gilthead sea bream (*Sparus aurata*) fed the experimental diets over 21 weeks.

	CV	CF	H10	H20	H40	P20	P40	H10P30	RC10	MA10	*p* Value	RMSE
TL, cm	25.13	25.41	25.96	26.09	25.94	25.52	25.77	26.30	25.67	25.20	NS	0.92
K, %	1.99	2.00	1.94	2.02	2.02	2.04	1.97	2.01	2.00	1.89	NS	0.11
FY, %	54.33	52.88	52.86	54.02	55.20	54.28	54.69	54.08	52.56	53.25	NS	3.09
HSI, %	0.95	1.04	0.92	1.04	1.02	1.00	0.88	1.08	0.96	0.87	NS	0.23
pH	6.17	6.19	6.16	6.15	6.21	6.14	6.13	6.24	6.20	6.21	NS	0.09
Texture, N	44.52	42.43	43.17	46.97	46.03	39.64	45.39	45.33	51.69	44.62	NS	10.65
**Skin colour**												
*L**	75.50 ^ab^	74.41 ^ab^	74.66 ^ab^	71.68 ^ab^	75.82 ^ab^	69.60 ^b^	73.99 ^ab^	73.01 ^ab^	70.87 ^ab^	76.41 ^a^	0.015	4.53
*a**	−2.69	−2.95	−2.81	−2.58	−2.78	−2.62	−2.32	−2.81	−2.55	−2.95	NS	0.55
*b**	−1.11 ^ab^	−3.85 ^b^	−0.35 ^ab^	−0.29 ^ab^	1.28 ^a^	−0.06 ^ab^	−1.86 ^b^	−1.46 ^b^	−1.27 ^ab^	−1.19 ^ab^	<0.0001	1.89
**Fillet colour**												
*L**	49.85	49.45	48.97	49.75	50.08	49.31	49.59	49.06	50.25	51.70	NS	2.15
*a**	0.30	0.31	−0.14	0.08	0.20	−0.11	0.06	0.29	0.10	−0.38	NS	0.68
*b**	−0.14	−0.97	−0.52	−2.14	−1.31	−0.62	−0.49	−0.49	−0.04	0.00	NS	1.92

CV, vegetable control; CF, fish meal control; H, *Hermetia illucens*; P, poultry by-product; RC, red swamp crayfish; MA, microalgae dried biomass. TL, total length; K, condition factor; FY, fillet yield; HSI, hepatosomatic index; VSI, viscerosomatic index; *L**, lightness; *a**, redness index; *b**, yellowness index. RMSE: root mean square error. ^a^, ^b^: Different superscript letters indicate significant difference among groups (*p* < 0.05). NS: not significant (*p* > 0.05).

**Table 5 animals-11-01919-t005:** Chemical composition (g 100 g^−1^ fresh tissue) and fatty acids profile (g 100 g^−1^ total FAME) of the fillets from gilthead sea bream (*Sparus aurata*) fed the experimental diets.

	CV	CF	H10	H20	H40	P20	P40	H10P30	RC10	MA10	*p* Value	RMSE
Moisture	69.31	69.66	69.85	68.87	69.36	69.23	69.99	69.27	69.13	70.00	NS	1.26
Ash	1.41	1.38	1.41	1.40	1.46	1.44	1.44	1.39	1.37	1.34	NS	0.11
Crude protein	19.74	19.85	19.93	19.91	19.78	20.22	20.16	20.30	20.29	19.58	NS	0.52
Total lipids	9.15	8.78	8.46	9.32	8.88	8.60	7.89	8.63	8.84	8.46	NS	1.46
**Fatty acids**												
C12:0	0.07 ^c^	0.29 ^bc^	0.59 ^b^	1.25 ^a^	1.61 ^a^	0.08 ^c^	0.12 ^c^	0.68 ^b^	0.14 ^c^	0.06 ^c^	<0.0001	0.28
C14:0	2.28 ^d^	3.32 ^a^	2.72 ^c^	2.92 ^bc^	3.07 ^ab^	2.24 ^d^	2.39 ^d^	2.65 ^cd^	2.46 ^cd^	2.31 ^d^	<0.0001	0.21
C16:0	14.61 ^b^	16.40 ^a^	14.82 ^b^	14.84 ^b^	14.76 ^b^	15.20 ^b^	15.17 ^b^	15.59 ^ab^	14.97 ^b^	14.54 ^b^	<0.0001	0.62
C16:1-n7	3.59 ^c^	5.35 ^a^	4.08 ^bc^	4.00 ^bc^	4.00 ^bc^	4.06 ^bc^	4.38 ^b^	4.51 ^b^	4.06 ^bc^	3.69 ^c^	<0.0001	0.36
C18:0	3.38 ^ab^	3.62 ^a^	3.24 ^ab^	3.11 ^b^	3.20 ^b^	3.51 ^ab^	3.53 ^ab^	3.46 ^ab^	3.34 ^ab^	3.20 ^b^	<0.0001	1.18
C18:1n-9	32.17 ^a^	27.22 ^b^	30.69 ^a^	31.26 ^a^	31.62 ^a^	32.14 ^a^	31.90 ^a^	31.49 ^a^	31.80 ^a^	30.79 ^a^	<0.0001	0.09
C18:1n-7	2.44 ^bc^	2.80 ^a^	2.49 ^bc^	2.42 ^c^	2.44 ^bc^	2.50 ^bc^	2.55 ^b^	2.51 ^bc^	2.54 ^bc^	2.52 ^bc^	<0.0001	1.28
C18:2n-6	16.33 ^ab^	10.22 ^c^	14.98 ^b^	15.09 ^b^	14.97 ^b^	15.39 ^b^	15.19 ^b^	14.68 ^b^	14.89 ^b^	17.99 ^a^	<0.0001	0.02
C18:3n-3	8.46 ^a^	7.63 ^b^	7.89 ^b^	7.92 ^b^	7.09 ^c^	7.43 ^bc^	6.72 ^c^	6.81 ^c^	7.96 ^ab^	8.37 ^a^	<0.0001	0.36
C20:1n-9	0.90 ^bc^	1.32 ^a^	1.02 ^b^	1.01 ^b^	0.95 ^bc^	0.93 ^bc^	0.96 ^bc^	0.93 ^bc^	0.95 ^bc^	0.86 ^c^	<0.0001	0.10
C20:5n-3	3.45 ^bc^	4.82 ^a^	3.87 ^b^	3.58 ^bc^	3.74 ^b^	3.64 ^bc^	3.76 ^b^	3.76 ^b^	3.79 ^b^	3.25 ^c^	<0.0001	0.33
C22:5n-3	1.21 ^c^	1.73 ^a^	1.42 ^b^	1.38 ^bc^	1.29 ^bc^	1.32 ^bc^	1.37 ^bc^	1.30 ^bc^	1.32 ^bc^	1.03 ^c^	<0.0001	0.13
C22:6n-3	5.17 ^bc^	7.72^a^	6.06 ^b^	5.41 ^bc^	5.51 ^bc^	5.52 ^bc^	5.75 ^bc^	5.48 ^bc^	5.67 ^bc^	4.94 ^c^	<0.0001	0.64
ƩSFA	21.18 ^c^	24.79 ^a^	22.26 ^bc^	22.92 ^bc^	23.46 ^ab^	21.86 ^bc^	22.04 ^bc^	23.22 ^b^	21.79 ^c^	20.89 ^c^	<0.0001	0.97
ƩMUFA	40.22 ^ab^	38.33 ^c^	39.47 ^b^	39.84 ^ab^	40.14 ^ab^	40.74 ^a^	40.94 ^a^	40.58 ^ab^	40.52 ^ab^	39.43 ^bc^	<0.0001	0.79
Ʃn-6 PUFA	17.70 ^ab^	11.57 ^c^	16.23 ^b^	16.26 ^b^	16.13 ^b^	16.87 ^b^	16.78 ^b^	16.19 ^b^	16.18 ^b^	19.34 ^a^	<0.0001	1.25
Ʃn-3 PUFA	19.91 ^bc^	23.86 ^a^	20.89 ^b^	19.87 ^bc^	19.18 ^bc^	19.48 ^bc^	19.12 ^c^	18.88 ^c^	20.39 ^bc^	19.42 ^bc^	<0.0001	1.24
n-3 PUFA/n-6PUFA	1.13 ^b^	2.16 ^a^	1.33 ^b^	1.22 ^b^	1.89 ^b^	1.15 ^b^	1.39 ^b^	1.17 ^b^	1.29 ^b^	1.00 ^b^	<0.0001	0.23

CV, vegetable control; CF, fish meal control; H, *Hermetia illucens*; P, poultry by-product; RC, red swamp crayfish; MA, microalgae dried biomass. SFA: saturated fatty acids; MUFA: monounsaturated fatty acids; PUFA: polyunsaturated fatty acids. The following FA were used for calculating the Ʃ classes of FAs but they are not listed because below 1% of total FAME: C13:0, C14:0, C14:1n-5, C15:0, C16:1n-9, C16:2n-4, C16:3n-4, C17:0, C17:1, C16:4n-1, C18:2n-4, C18:3n-6, C18:3n-4, C18:4n-3, C18:4n-1, C20:0, C20:1n-11, C20:1n-7, C20:2n-6, C20:3n-6, C20:4n-6, C20:3n-3, C20:4n-3, C21:5n-3, C22:0, C22:1n-9, C22:1n-11, C22:1n7, C22:2n-6, C22:4n-6, C22:5n-6, C24:0. RMSE: root mean square error. ^a,b,c^: different superscript letters indicate significant difference among groups (*p* < 0.05). NS: not significant (*p* > 0.05).

**Table 6 animals-11-01919-t006:** Conjugated dienes (CD, mmol kg^−1^ fresh tissue) and thiobarbituric acid reactive substances (TBARS, mg MDA-eq. kg^−1^ fresh tissue) of the fillets from gilthead sea bream (*Sparus aurata*) fed the experimental diets.

	CV	CF	H10	H20	H40	P20	P40	H10P30	RC10	MA10	*p* Value	RMSE
CD	0.20 ^b^	0.27 ^a^	0.22 ^ab^	0.25 ^ab^	0.23 ^ab^	0.21 ^ab^	0.21 ^ab^	0.22 ^ab^	0.24 ^ab^	0.20 ^b^	0.014	0.04
TBARS	0.86	1.22	0.71	0.76	0.82	0.77	0.65	0.72	0.70	0.74	NS	0.37

CV, vegetable control; CF, fish meal control; H, *Hermetia illucens*; P, poultry by-product; RC, red swamp crayfish; MA, microalgae dried biomass. RMSE: root mean square error. ^a^, ^b^: different superscript letters indicate significant difference among groups (*p* < 0.05). NS: not significant (*p* > 0.05).

## Data Availability

Not available.

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
