# Peer review of "Appetite Regulation, Growth Performances and Fish Quality Are Modulated by Alternative Dietary Protein Ingredients in Gilthead Sea Bream (Sparus aurata) Culture"

_animals, 2021, doi:10.3390/ani11071919_

Round 1
Reviewer 1 Report
In General
This paper describes the possibility to replace partially dietary protein ingredients by alternative sources. Parameters determined on cultured fish (Sparus aurata) were so many and analyses were so laborious. The authors concluded that up to 40 % replacement of insect (Hermetia illucens) prepupae, poultry-by products, and red swamp crayfish (Procambarus clarkii) did not show any detrimental effects on the growth performances, gene expression, and the quality of fillets.
Aims of the study is very important because these alternative dietary protein sources are a “waste” (PBM) or organisms growing on waster food (H), both of which are not directly destinated to human consumption. When we consider it will be difficult to obtain enough amount of fishmeal in the future, such studies as the present one are more and more valuable. Context of the study is clear and discussion is appropriate. Therefore, a referee considers this article is valuable to be published.
However, there are so many abbreviations throughout the text, and a referee must confirm what the abbreviation means so often. A referee suggests the authors to show full text of each abbreviation more frequently in the text.
Specific Comments
- Introduction (p. 2, line 69)
“FM” must be changed to “FM (fish meal)”.
- Materials and Methods (p. 4, l. 150)
Abbreviation of poultry by-product meal should be uniform as “P” or “PBM”.
- Results (p. 7, Table 3)
FW should be expressed in full (final weight?).
- Discussion (p. 12, from l. 439 to p. 13, l. 502)
Discussion on the PUFAn-3 is too long and boring. A referee suggests the authors to make it much more compact and to focus on the most important point.
Author Response
The detailed answers to the requests of Rev#1 are in the attached file

Reviewer 2 Report
The authors present an interesting study on the use of several dietary formulations for the cultivation of the gilthead seabream (Sparus aurata). The study is sound and provides interesting new insights on the use of alternative ingredients for formulated diets in fish farming. The manuscript is nicely presented and organized. The content is clear and easily readable, and the experimental design and methodologies applied are exhaustive, detailed and adequate. Finally, the results achieved are promising. I consider that the manuscript deserves publication after minor changes/comments are properly addressed.
I have a few comments on the document:
Summary
L15-20: Please, synthesize this text.
Introduction
L73 – Use n-3 PUFA instead of PUFAn-3. Also, check and correct in Table 5 and along the manuscript.
Materials and Methods
The experimental design is well explained and appropriate. The analyses performed are perfectly described and precise.
Results
L296-298. Use the same font size for this part. Also in Tables 5 and 6.
Discussion
L398-403 and L522-525: Check and modify font size.
Can the authors roughly estimate feasibility (considering source availability of the proposed ingredients) and economic profits using the proposed dietary ingredients compared to conventional formulated diets?
Tables 4 and 5: Edit column wides properly.
Author Response
The detailed answers to the requests of Rev#21 are in the attached file
